# Lung Organoids as Model to Study SARS-CoV-2 Infection

**DOI:** 10.3390/cells11172758

**Published:** 2022-09-04

**Authors:** Li Peng, Li Gao, Xinya Wu, Yuxin Fan, Meixiao Liu, Jingjing Chen, Jieqin Song, Jing Kong, Yan Dong, Bingxue Li, Aihua Liu, Fukai Bao

**Affiliations:** 1The Institute for Tropical Medicine, Faculty of Basic Medical Science, Kunming Medical University, Kunming 650500, China; 2Yunnan Health Cell Biotechnology LTD, Kunming 650031, China; 3Yunnan Province Key Laboratory of Children’s Major Diseases Research, The Affiliated Children Hospital, Kunming Medical University, Kunming 650030, China

**Keywords:** organoid, lung, coronavirus, COVID-19, SARS-CoV-2

## Abstract

Coronavirus disease-2019 (COVID-19), caused by severe acute respiratory syndrome coronavirus 2 (SARS-CoV-2), has become a global pandemic and has severely affected socio-economic conditions and people’s life. The lung is the major target organ infected and (seriously) damaged by SARS-CoV-2, so a comprehensive understanding of the virus and the mechanism of infection are the first choices to overcome COVID-19. Recent studies have demonstrated the enormous value of human organoids as platforms for virological research, making them an ideal tool for researching host–pathogen interactions. In this study, the various existing lung organoids and their identification biomarkers and applications are summarized. At the same time, the seven coronaviruses currently capable of infecting humans are outlined. Finally, a detailed summary of existing studies on SARS-CoV-2 using lung organoids is provided and includes pathogenesis, drug development, and precision treatment. This review highlights the value of lung organoids in studying SARS-CoV-2 infection, bringing hope that research will alleviate COVID-19-associated lung infections.

## 1. Introduction

In December 2019, the first case of a new coronavirus, Coronavirus disease 2019 (COVID-19), was reported in Hubei Province, China [1]. Over the next 1 to 2 years, the disease spread worldwide. To date, the World Health Organization (WHO) has reported more than 510 million confirmed cases and 6.26 million deaths worldwide [2]. COVID-19 has triggered a global pandemic. In recent years, various mutants have appeared one after another [3,4], which makes epidemic prevention and control more difficult. Although some studies have shown that several drugs can improve the clinical symptoms of COVID-19 [5], the efficacy of their treatment is controversial [6]. The main protective measures currently in place are vaccinations, masks, and isolation. Unfortunately, new infections occur every day. Furthermore, people’s work and life have been seriously affected. It is urgent to study this new coronavirus and find ways to eradicate this pandemic as soon as possible.

COVID-19 is caused by the severe acute respiratory syndrome coronavirus 2 (SARS-CoV-2) [7], which belongs to the SARS-like virus cluster. It has 86% homology at a nucleotide level with the first detected SARS-CoV [8]. The clinical manifestations range from asymptomatic to mild respiratory infection and influenza-like diseases (mainly fever, cough, and fatigue) to life-threatening diseases accompanied by lung injury, multiple organ failure, and death [7,8]. Faced with this serious epidemic, scholars have devoted themselves to studying the mechanism of viral infection, pathogenic mechanisms, and the development of effective therapeutic drugs to tackle COVID-19 in its early stages. Scientists use immortalized cell lines from humans and animals, including Vero E6, Calu-3, and Caco-2, to study cellular susceptibility, infection mechanism, and the antiviral drug screening of the virus [9,10]. These cell lines have the advantages of convenience and easy gene editing, but they cannot represent the cell state in the actual tissue because of the single cell type [11]. In addition, SARS-CoV-2 uses its spike (S) protein to enter host cells by binding to angiotensin-converting enzyme 2 (ACE2) on the host cell surface. Thus, cells expressing ACE2 are susceptible to SARS-CoV-2 infection. Accordingly, SARS-CoV-2 has high infectivity in primates, including humans, but low infectivity in wild-type mice, limiting animal species that can be used for experiments. Moreover, it is ethically difficult to use these animals in large quantities [12]. Therefore, physiologically relevant human embryonic stem (ES)/induced pluripotent stem (iPS) cell-derived organoids are undoubtedly a better model for studying SARS-CoV-2 infection. Since the lung is the major target organ infected and (seriously) damaged by SARS-CoV-2, uncovering the mechanism of lung infection is the key to controlling COVID-19. In this review, existing data about SARS-CoV-2 infection using lung organoids are summarized.

## 2. Lung Organoids

The term “organoid” first appeared in the 1940s in reference to a tumor case study [13]. Initially, organoids were not clearly defined as this term has often been used to refer to cultures or structures that resemble an organ [14]. Decades later, several groups attempted to define organoids as stem-cell-derived or organ progenitor-derived multicellular systems in which cells spontaneously self-organize and self-renew into properly differentiated, functional cell types [15,16]. Organoids are usually cultured in a three-dimensional (3D) system containing a combination of growth factors and nutrients. Derivation methods differ according to the type of organ, such as intestinal, kidney, brain, and lung. These 3D structures can be either embedded within an extracellular hydrogel matrix, such as Matrigel, or cultured in submerged culture conditions [15,17,18]. Clevers et al. created the first organoids derived from adult stem cells, which grow almost indefinitely under the right conditions, and can build themselves into complex structures that reflect their organ origin [19]. Organoids were previously used mostly to study basic human biology and development, with only a few groups using the models to study viruses and other infectious diseases [20]. However, with the advancement of culture technology and the advantages of organoid models, organoids have become a hotspot in scientific research. The pandemic has especially brought the use of organoids to center stage.

Lung organoids can be obtained from both primary cell culture and stepwise induction of pluripotent stem cells. In the first method, alveolar organoids were successfully generated from single adult human alveolar epithelial type II (AT2) or KRT5+ basal cells [21]. Furthermore, airway organoids were generated from the human primary airway epithelium [22]. In the second method, lung organoids can be established from multifarious human stem cells (hSCs), including pluripotent stem cells (iPSCs), embryonic stem cells (ESCs), and adult or fetal stem cells derived from surgical specimens. The culture of lung organoids includes four definitive periods: (1) definitive endoderm (DE), (2) anterior foregut endoderm (AFE), (3) lung progenitor cells (LPCs), and (4) all types of lung organoids [17,23,24]. The type of organoid depends on the induction strategies and types of reagents in each period. DE is one of the three germ layers of the embryo proper and is a prerequisite for lung development. The lung originates from lung buds that arise on the anterior ventral aspect of the DE and develop into the lung and corresponding airways through a complex and coordinated process of branching morphogenesis and lineage specification [25]. Differentiation of hSCs in the presence of activin A and low serum or serum-free media [26] produced cultures consisting of up to 80% definitive endoderm cells. This population was further enriched to near homogeneity using the cell-surface chemokine receptor, CXC motif chemokine receptor type 4(CXCR4) [27]. Subsequently, dual inhibition of transforming growth factor (TGF)-β and bone morphogenic protein (BMP) signaling after specification of definitive endoderm from hSCs resulted in a highly enriched AFE population. In several experiments, Green et al. [28] demonstrated that only in the combined utilization of NOGGIN, a physiological inhibitor of bone morphogenic protein (BMP) signaling, and SB-431542, a pharmacological inhibitor of activin A/nodal and transforming growth factor beta (TGF-β) signaling, was AFE specific. AFE is then ventralized after signaling by the WNT, BMP, fibroblast growth factor (FGF), and retinoic acid (RA) pathways are activated to obtain lung progenitor cells (LPCs). The early LPCs can differentiate into lung and airway epithelial cells in vivo after transplantation under the kidney capsule of immunodeficient mice [29]. Meanwhile, early LPCs can also differentiate into predominantly distal cells in vitro [26]. Some scholars sort and purify the LPCs at this stage and make “epithelial only” lung organoids [30,31], while others use completely formed spheres germinated from monolayer cultures [29,32]. Subsequently, LPCs are differentiated into specific types of lung organoids, including alveoli [30,33], proximal airways [34,35], or a mixture of both [26,36] in a 3D culture system. However, hSCs-derived lung organoids failed to become “mature” in vitro (such as less differentiated, and their gene activity resembles that of the lung of a developing human fetus). Spence et al. also used RNA-sequencing to compare the global transcriptional profile of organoids to human fetal and adult lungs, undifferentiated ESCs, and definitive endoderm. It turned out that these organoids had a striking similarity to human fetal lungs [32,37]. Interestingly, fetal-like lung organoids will serve as an unparalleled model for studying human lung development and ultimately allow us to develop new treatments for premature infants with immature lungs at birth [38]. The various existing lung organoids and identification biomarkers are summarized in addition to their applications and limitations to assist the wider use of lung organoids in the future (Table 1). These lung organoids have a near-physiological structure, have retained the functions of the original tissue, and provide a powerful platform for SARS-CoV-2 studies.

## 3. SARS-CoV-2

The name “coronavirus (CoVs),” coined in 1968, is derived from the “corona”-like morphology observed for these viruses in the electron microscope [44]. To date, seven CoVs have been reported to infect humans (Figure 1), of which HCoV-229E and HCoV-OC43 have been shown to cause common cold-like illnesses. They are also the first coronavirus found to infect humans [45,46]. Severe acute respiratory syndrome coronavirus (SARS-CoV), a Beta-coronavirus, emerged in late 2002 in Guangdong province, China, and spread rapidly to other countries and continents [47]. The species of bats are a natural host of this coronavirus [48]. Meanwhile, SARS-CoV, which causes severe acute respiratory syndrome (SARS), has introduced a new chapter in the serious human disease caused by coronavirus [44]. Subsequently, reports of two new human coronaviruses, (HCoV-NL63 in 2004 [49] and HCoV-HKU1 in 2005 [50]) associated with respiratory disease have been published. Several years later, the Middle East respiratory syndrome coronavirus (MERS-CoV) emerged in 2012 and caused outbreaks in Saudi Arabia and South Korea [51]. Similar to SARS-CoV, MERS-CoV originated in bats, but dromedary camels were identified as intermediate hosts [52]. Taken together, the repeated occurrence of coronaviruses in humans and the detection of numerous coronaviruses in bats suggest that future zoonotic transmission events are likely to continue [53]. Indeed, 2019 saw the introduction of a novel zoonotic coronavirus in Hubei Province, China, which appears to be a close relative of the SARS-CoV and has been named SARS-CoV-2. It is a non-segmented, enveloped, positive-sense single-stranded RNA (ssRNA) virus that belongs to the Beta-coronavirus genus of the Coronaviridae family and can infect both humans and animals [8]. The disease caused by this virus was named coronavirus disease 2019 (COVID-19) [1]. The virus has rapidly spread worldwide, leading the WHO to declare COVID-19 a pandemic in March 2020 [54].

The particles of coronaviruses are spherical with three viral proteins anchored in the envelope: (1) the triple-spanning membrane (M) protein, (2) the envelope (E) protein, and (3) the spike (S) protein, which forms the characteristic trimeric spikes [23,24,55]. S consists of two functional subunits responsible for binding to the host cell receptor (S1 subunit) and fusion of the viral and cellular membranes (S2 subunit), which contain 666 and 583 amino acid residues, respectively (Figure 2). In the case of several avian and mammalian coronaviruses, S is cleaved by Furin or a related protease into the S1 and S2 subunit. Region S1 contains NTD and receptor-binding domain (RBD). SARS-CoV-2 and other human coronaviruses, SARS-CoV and HCoV-NL63, use as their receptor a cell-surface zinc peptidase, angiotensin-converting enzyme 2 (ACE2) [12,56]. Compared with SAR-CoV, the receptor-binding domain (RBD) of SARS-CoV-2 binds to the ACE2receptor with higher affinity and stability [57]. The binding of the receptor-binding motif (RBM) within the RBD to ACE2 mediates membrane fusion and viral entry. It is the critical determinant of viral host range and tropism [54]. Mutations within both the RBDs of S1 and S2 have been shown to influence pathogenesis [53,58]. Region S2 contains two heptad repeat regions (HR1 and HR2) involved in peptide synthesis in addition to a fusion peptide (FP), transmembrane anchor (TA), and intracellular tail (IT) associated with fusion activity [58]. Therefore, the viral S protein and the way that the virus uses the S protein to enter the host cells are the first targets for vaccine development and overcoming COVID-19.

## 4. Studies on SARS-CoV-2 with Lung Organoids

### 4.1. Clarifying the Pathogenesis of COVID-19

The pathogenesis of COVID-19 is largely unknown, and the main causes of death are cardiopulmonary failure and coagulation dysfunction [59]. Lung autopsies show diffuse alveolar damage, endothelial injury, extensive thrombosis [60], and increased angiogenesis [61,62]. Viral genome sequences have also been detected in multiple organs and tissues, including lungs, heart, gut, kidneys, pharynx, and brains of COVID-19 patients [63,64]. These findings suggest that studying the cell tropism of SARS-CoV-2 and how it enters cells is the first step in clarifying COVID-19 pathogenesis (Figure 3).

SARS-CoV-2 cell tropism of different sources and types of lung organoids varies. The type(s) of cells that are infected by the virus is still controversial. SARS-CoV-2 was found to only infect AT2 cells in lung organoids developed by iPSCs [23]. However, the virus infected ciliated, club, and AT2 cells in ESC-derived lung organoids, including airway and alveolar [42]. To facilitate better contact of the organoids with the virus, the organoids can be cultured using the 2D air-liquid interface (2D-AL1) culture system or “flipped” to form a state with the top pointing in an outward direction. Alveolar, basal, and rare neuroendocrine cells, which are grown from fetal lung bud tip organoids, were readily infected by SARS-CoV-2, especially AT2 cells [65]. SARS-CoV-2 is more likely to infect the upper respiratory tract than the distal lung [66,67]. This may be related to the distribution of ACE2 and TMPRSS2, which are receptors and helpers for viral entry into cells, respectively. In the lungs, ACE2 and TMPRSS2 are mainly expressed in ciliated bronchial cells and to a lower level in AT2 cells but not in differentiated AT1 cells [66,67,68]

The cell entry of SARS-CoV-2 depends on the binding of the viral S protein to cellular receptors ACE2 and S protein priming by host cell proteases, such as TMPRSS2 [10]. Recent studies have shown that SARS-CoV-2 also utilizes cellular proteases, such as cathepsins B and L (CTSB and CTSL, respectively), to enter host cells [69]. Moreover, ACE2 interacts with the SARS-CoV-2 S protein with sufficient affinity to cause human transmission [57].

### 4.2. COVID-19 Drug Development

The COVID-19 pandemic has been ongoing for more than 2 years. For this pandemic, it will take a relatively long time to achieve herd immunity through vaccination, and vaccination alone cannot completely block viral invasion. Therefore, both domestic and foreign countries have been actively exploring therapeutic drugs. At present, this exploration mainly focuses on the expansion of the indications of the listed drugs [42], the screening of drugs in the clinical trial stage, and the research and development of new drugs [70]. Some researchers classify effective drugs based on the viral infection process, including the fusion of the virus with the plasma membrane/endosomal membrane into target cells, viral replication and translation, new virus assembly and release, and anti-inflammatory aspects [71]. Some researchers classify different phases of viral infection, including initial infection followed by pulmonary and inflammatory phases [72]. At present, many effective antiviral drugs or compounds have been screened, but the effects are organ-heterogeneous, and the antiviral effects are controversial. In addition, as the main site of COVID-19 infection in the body is the lung, only a few drugs have been tested on lung organoids, and these studies mainly focus on the entry and replication of the virus (Table 2).

Remdesivir, an RNA-dependent RNA polymerase inhibitor, can shorten the recovery period of patients and prevent the progression to more severe respiratory disease by inhibiting viral replication [80]. This drug causes a reduction in the production of infectious viral particles in human proximal airway organoids (hAWOs) and alveolar organoids (hALOs) [42]. On May 1, 2020, the United States Food and Drug Administration issued an emergency use authorization for Remdesivir treatment in adults and children hospitalized with suspected or laboratory-confirmed COVID-19 [80]. However, the antiviral efficacy of Remdesivir against SARS-CoV-2 is still controversial. A recent randomized phase 3 clinical trial for SARS-CoV-2 infection showed that Remdesivir was ineffective [81], suggesting that its effective treatment may be limited to specific groups. Camostat, an inhibitor of the serine protease TMPRSS2, was found to partially block the viral S protein entry into cells [10,82]. Camostat showed a slightly inhibitory effect in hAWOs but not in hALOs [42]. Other drugs, such as Dutasteride, Finasteride, and Ketoconazole, lead to a decrease in the ACE2 levels in hALOs and efficiently prevent viral infection [73,75]. However, studies using organoid models also have ruled out some drugs or compounds that were ineffective against lung infections. Enzalutamide, a potent inhibitor of the androgen receptor (AR) [83], may have antiviral activity in the prostate glands of male COVID-19 patients but has no therapeutic effect in patients with a COVID-19 lung infection. Studies have shown that Enzalutamide could efficiently prevent SARS-CoV-2-driven entry into prostate cells by inhibiting AR, leading to a reduction in TMPRSS2 expression, but without inhibiting infection of hLOs with SARS-CoV-2 [74]. Overall, these studies demonstrate the benefits of human lung organoids in the search for anti-SARS-CoV-2 drugs (Figure 3). Furthermore, organoids can evaluate the toxicity and safety of COVID-19 therapeutics [12].

### 4.3. Promoting Precision Treatment of COVID-19

Since human iPSCs can be established from individuals of any genetic background, lung organoids derived from iPSCs or adult stem cells (ASCs) represent a powerful model to study the relationship between individual differences and COVID-19 severity and, when coupled with the application of gene editing, will help achieve precision treatment for COVID-19 patients (Figure 3).

Studies have shown that gender differences in the infection efficiency of SARS-CoV-2 exist [12]. Among them, the mortality rate of men is higher than that of women, especially older men [84,85]. The elderly population is a specific group of COVID-19 patients with severe disease and unique symptoms, predominantly isolated confusion, falls, absence of fever, and/or digestive manifestations [86]. In addition to age and gender, comorbidities have also been identified as major risk factors for severe forms and death in patients with COVID-19 [87,88]. Hypertension was the most common comorbidity, followed by cardiovascular disease, diabetes, and hypercholesterolemia. In an analysis of baseline characteristics and outcomes of 1591 patients with COVID-19, of 1043 patients with available data, 709 had at least one comorbidity and 509 had hypertension [89]. These results demonstrate that COVID-19 severity is closely related to individual differences. ASCs-derived organoids can demonstrate individual differences and can be used to compare the viral responses of people of different ages, genders, races [90], and underlying diseases to obtain personalized treatment plans.

When combined with clustered, regularly interspaced short palindromic repeats (CRISPR), organoids may answer many unanswered questions related to SARS-CoV-2. Precise gene editing to simulate allelic variation in organoids through CRISPR can be used to screen for SARS-CoV-2 susceptibility genes. In addition, CRISPR can also identify the role of host proteins or signaling pathways in SARS-CoV-2-mediated pathophysiology through gene knockout studies or by utilizing CRISPRa/I, which allows gene activation or repression [91]. Using CRISPR, Dobrindt et al. demonstrated that a single nucleotide polymorphism (SNP) in the furin gene is associated with SARS-CoV-2 infection in human iPSCs [92]. Using this technology, Wang et al. obtained different isoforms of apolipoprotein E (ApoE), showing that isoforms affect COVID-19 aggregation [93]. Therefore, the full use of 3D organoid models combined with various technologies and research platforms can effectively promote the clinical translation of basic science and achieve precise treatment for COVID-19.

## 5. Conclusions and Future Perspectives

Compared with 2D cells and animal models, 3D lung organoids show great potential in the study of SARS-CoV-2. This 3D organoid can not only mimic the natural cellular microenvironment and cell–cell interactions but also exhibit host–pathogen interactions caused by viral infections. Such an organoid provides a reliable platform to study the pathogenesis of viral infection for drug candidate development and personalized medicine. However, lung organoid models also have certain limitations. First, lung organoids fail to become “mature” in vitro. Human iPS/ES-derived lung organoids are closer to a fetal state than an adult state. Interestingly, transplantation of organoids into the kidney subcapsular or epididymal fat pad of mice resulted in significant morphological and functional maturation into the adult state [38]. Further work is required to understand this phenomenon better. Second, the human body is a multi-organ, multi-system organism. Lung organoids that are currently cultured, while mimicking the structure and multicellular function of specific tissues in the body, do not represent typical environments that are usually found in tissues due to their lack of immune, circulatory, and nervous systems. Interactions between organs cannot be modeled either. This limitation is also a technical barrier that needs to be solved immediately. Another limitation is that current organoid culturing is time-consuming and expensive. Various problems arise during the culture process, including different media components, non-uniform sizes, and differences in the degree of differentiation. These issues prevent extensive and robust use for disease modeling. Researchers need to streamline the production process and standardize cultural methods to create thousands of uniform organoids rapidly and cheaply.

Despite these challenges, the application potential of lung organoids is enormous. The vascular organization of future organoids is a major challenge, but researchers have applied embedded 3D bioprinting to introduce perfusable vascular channels into 3D organoid tissue, thus creating a perfusable cardiac tissue [94]. The research holds promise for the fusion of lung organoids with the circulatory system. Another successful study describes the generation of intestinal tissue with an enteric nervous system by combining human intestinal organoids and pluripotent stem cell-derived neural crest cells for recapitulating enteric nervous system development [95]. These examples demonstrate that it is feasible to grow multi-system, multi-layered organoids in vitro, which enables organoids to more realistically mimic the extra-epithelial microenvironment in human disease models and pharmacokinetic studies and can accelerate the discovery of new drugs against SARS-CoV-2. In addition, artificial intelligence (AI) has recently gained attention for accelerating the repurposing of drugs for several challenging diseases, including COVID-19 [96,97]. It is suggested that defeating COVID-19 as soon as possible requires the combined use of multiple technologies and multiple platforms. The combination of the microfluidic device and organoid models can effectively study the crosstalk between different organs, which cannot be done by the organoid model alone [98]. Computational platforms for drug screening can rapidly screen drug candidates for efficacy against COVID-19. At the same time, organoid models are also needed to study the efficacy and toxicity of these drugs [91]. Large-scale current good manufacturing practice (cGMP) grade production of organoids enables the future manufacture of “transplantable” organoids and tissues, opening up the potential of organoids as advanced therapy medicinal products (ATMPs) [99]. Currently, multi-organoid systems have also been developed in which each organoid is located within specific compartments that are interconnected by microchannels and mimic the body’s systemic blood circulation, organ proximity, and function [91]. Multiple organ failure is a typical occurrence in severe COVID-19 patients. Using multi-organoid systems can effectively allow the study of the effects of one infected cell in a patient on one or more surrounding cells. While isolating tissue samples and establishing lung organoids from deceased COVID-19 patients is challenging given the infectivity of SARS-CoV-2, this provides insight into virus-host interactions and may shed light on new therapeutic mechanisms for COVID-19. Overall, lung organoids provide critical complementary information in SARS-CoV-2 infection research and can aid the discovery of new targets that could accelerate the search for effective therapies and prevention of COVID-19.

## Figures and Tables

**Figure 1 cells-11-02758-f001:**
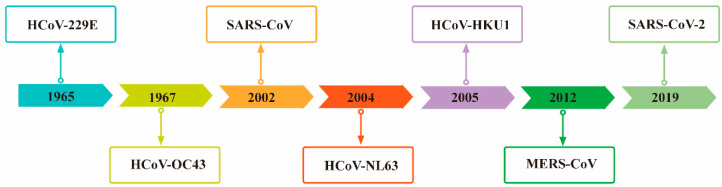
Chronological order of seven coronaviruses capable of infecting humans.

**Figure 2 cells-11-02758-f002:**
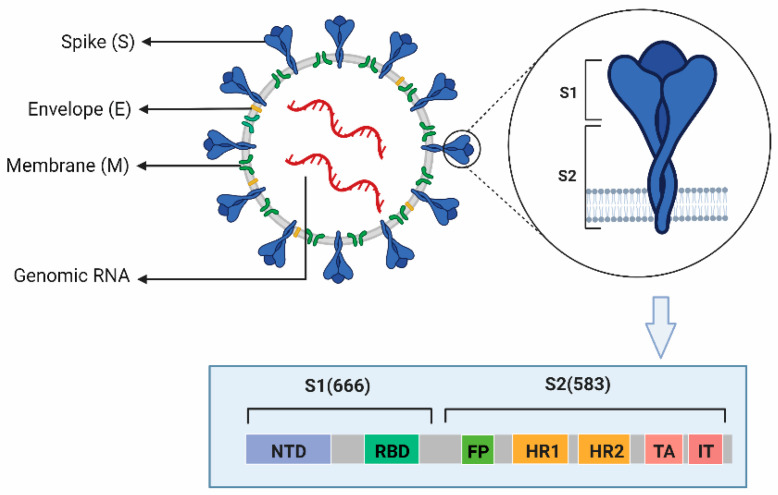
The structure of SARS-CoV-2. (This Figure was created with BioRender software. https://biorender.com/).

**Figure 3 cells-11-02758-f003:**
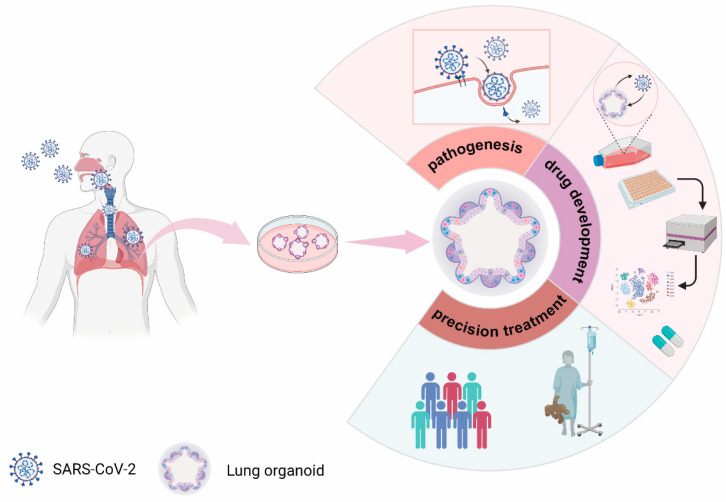
Current study of SARS-CoV-2 using lung organoids.(This Figure was created with BioRender software. https://biorender.com/).

**Table 1 cells-11-02758-t001:** Identification and application of existing lung organoids.

Organoid Model	Original Cells or Tissues	Differentiated Cell Types and Biomarker	Potential Application	Limitation	Ref.
Human lung organoids (HLOs)	hPSCs iPSCs	Goblet cells (MUC5AC) Ciliated cells (FOXJ1, ACTTUB) Multi-ciliated cells (FOXJ1) Club cells (SCGB1A1) Basal cells (P63, NGFR, CK5) ATII cells (SFTPC, SFTPB) ATI cells (HOPX, PDPN) Distal progenitor (NMYC, ID2) Proximal lung (SOX2) Distal lung (SOX9)	Lung development, Proximal and distal epithelium maturation, Epithelial–mesenchymal interactions, Airway remodeling after injury, Epithelial–mesenchymal crosstalk.	No branching morphogenesis; Lack of immune cells, vasculature, innervation.	[32] [37] [39] [40]
Lung bud organoids (LBOs)	hPSCs hESCs	Goblet cells (MUC5AC+, MUC5B) Ciliated cells (FOXJ1) Club cells (CC10) Basal cells (p63+) ATII cells (SFTPC, SFTPB, ABCA3, HT2-280) ATI cells (HT1-56, HOPX, PDPN, CAV1, SCNN1A, AKAP5, CLIC5) Neuroendocrine cells (SNY, CHGA)	Lung development, Epithelial fate decisions, Pulmonary fibrosis.	Terminal maturation, Branching appears random, Exact nature and patterning of the mechanism unclear, Biased toward distal lung.	[17] [41] [39]
Human proximal airway organoids (hAWOs)	hPSCs	Multi-ciliated cells (FOXJ1) Goblet cells (MUC5AC+) Basal cells (TP63+, KRT5+) Club cells (SCGB3A2+)	Inherited airway diseases (cystic fibrosis, primary ciliary dyskinesia), Drug screening, Precision medicine.		[35] [42]
Human alveolar organoids (hALOs)	hPSCs lung epithelial stem lung progenitor cells	ATII cells (SFTPC) ATI cells (PDPN)	Respiratory diseases (idiopathic pulmonary fibrosis, tuberculosis infection, respiratory virus infection), Individualized medicine.	Hard to model inflate and deflate during gas exchange; Lack of developed vasculature.	[43] [42]

**Table 2 cells-11-02758-t002:** COVID-19 drug research on lung organoids.

Drugs/Compounds	Organoids Model	Brief Description	Ref.
Remdesivir	Human airway organoids, Human alveolar organoids	Inhibit virus replication and infection, Inhibit virus replication	[73] [42]
Camostat	Human airway organoids, Human lung organoids, Human alveolar organoids	Inhibit virus infection, No inhibition effects	[74] [24] [42]
Dutasteride	Human alveolar organoids	Decrease ACE2 levels and inhibit virus infection	[73]
Finasteride	Human alveolar organoids	Decrease ACE2 levels and inhibit virus infection	[73]
Ketoconazole	Human alveolar organoids	Decrease ACE2 levels and inhibit virus infection	[75]
Bestatin	Human alveolar organoids; Human airway organoids	No inhibition effects	[42]
Imatinib	Human alveolar organoids	Inhibit virus entry	[23]
Nafamostat	Human alveolar organoids	Inhibit virus infection	[24]
Quinacrine dihydrochloride (QNHC)	Human alveolar organoids	Inhibit virus infection	[23]
Mycophenolic acid (MPA)	Human alveolar organoids	Inhibit virus infection	[23]
25-hydrocholesterol	Human alveolar organoids	Inhibit virus entry	[76]
Hydroxychloroquine	Human alveolar organoids	Inhibits virus replication and infection	[73]
EK1 peptide	Human alveolar organoids	Inhibits virus infection	[24]
Aloxistatin (E-64d)	Human alveolar organoids Human airway organoids	No inhibition effects	[77,78]
EIDD-2801	Human alveolar organoids; Human airway organoids	Inhibit virus replication and infection	[42,79]
Neutralizing antibodies (C86)	Human alveolar organoids; Human airway organoids	Inhibit virus replication	[42]
Humanized COVID-19 decoy antibody	Human airway organoids	Inhibit virus entry and infection	[70]
IFNB1	Human alveolar organoids, Human airway organoids	Inhibits virus replication and infection, No inhibition effects	[73]
EIDD-2801	Human alveolar organoids; Human airway organoids	Inhibit virus replication and infection	[42,79]
IFNα	Human alveolar organoids	Inhibits virus replication	[68]
IFNγ	Human alveolar organoids	Inhibits virus replication	[68]
IFN-λ1	Human airway organoids	Inhibits virus replication and infection	[65]

## Data Availability

Data sharing does not apply to this article as no new data were created or analyzed in this study.

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
