# Peer review of "Lung Organoids as Model to Study SARS-CoV-2 Infection"

_cells, 2022, doi:10.3390/cells11172758_

Round 1

Reviewer 1 Report

The manuscript about the lung organoid models is overall well written and thesubject is timely.

I have several comments.

In the abstract and also in the text (Line 63-) , the authors described that lung is the major target organ infected by SARS-Cov2. However, the authors also described the virus genome was found in various organs (Line 177-). Does this mean the virus infect various organs but much higher amount is found in the lung amongst of all? I think that various organs are infected but seriously damaged in the lung amongst of all. 

In the Conclusion, the authors described that the lung organoids fail to become mature. However, the authors did not explain how immature in the main text (lung organoid section). It is better to explain that in the main text with appropriate references.

Minor, in Line 100, the authors should fully spell CXCR4.

Line 161, full spell of ACE2 should be before ACE2 in the Line 160. 

Line 219, hAWOs and hALOs should be fully spelled before the abbreviations.

Author Response

Response to Reviewer 1 Comments (Please see the attachment)

To reviewer #1: We appreciate your careful reminding. Your comments are very valuable and helpful for revising our paper. All changes had been made with red colored text in the current manuscript and our itemized responses to your comments, questions, and suggestions (repeated below for your convenience) are as follows:

Point 1: In the abstract and also in the text (Line 63-), the authors described that lung is the major target organ infected by SARS-Cov2. However, the authors also described the virus genome was found in various organs (Line 177-). Does this mean the virus infect various organs but much higher amount is found in the lung amongst of all? I think that various organs are infected but seriously damaged in the lung amongst of all.

Response 1: Thank you for your comment. Your understanding is correct, multiple authoritative articles reported that COVID-19 patients mainly died of progressive respiratory failure, and the lungs were the main damaged organ. In addition, the results of SARS-CoV-2 viral load in autopsy tissue samples obtained from 22 patients who had died from COVID-19 showed that the highest levels of SARS-CoV-2 copies per cell were detected in the respiratory tract, and lower levels were detected in the other organs.

(Ackermann M, Verleden SE, Kuehnel M, et al. Pulmonary Vascular Endothelialitis, Thrombosis, and Angiogenesis in Covid-19. N Engl J Med. 2020. 383(2): 120-128.

Puelles VG, Lütgehetmann M, Lindenmeyer MT, et al. Multiorgan and Renal Tropism of SARS-CoV-2. N Engl J Med. 2020. 383(6): 590-592.)

Point 2: In the Conclusion, the authors described that the lung organoids fail to become mature. However, the authors did not explain how immature in the main text (lung organoid section). It is better to explain that in the main text with appropriate references.

Response 2: Thank you for your advice. Your suggestion is very valuable and helpful. We have explained the immaturity of lung organoids in Table 1, such as no branched morphology, etc., and at the same time, we have added a detailed description in the main text with appropriate references, the following parts marked in red are the added sentences.

“However, hSCs-derived lung organoids failed to become “mature” in vitro (such as less differentiated, and their gene activity resembles that of the lung of a developing human fetus). Spence et al. also used RNA-sequencing to compare the global transcriptional profile of organoids to human fetal and adult lungs, undifferentiated ESCs, and definitive endoderm. It turned out that these organoids had a striking similarity to human fetal lungs [32,37]. Interestingly, fetal-like lung organoids will serve as an unparalleled model for the study of human lung development, and ultimately allow us to develop new treatments for premature infants with immature lungs at birth [93].”

Point 3: Some minor errors are in the manuscript, such as:

In Line 100, the authors should fully spell CXCR4.

Line 161, full spell of ACE2 should be before ACE2 in the Line 160.

Line 219, hAWOs and hALOs should be fully spelled before the abbreviations.

Response 3: Thank you for your advice. We’ve checked the written errors throughout the manuscript and corrected the mistakes found in the revised manuscript.

Reviewer 2 Report

In this review-manuscript, Peng et al extensively described the current status and the importance of human lung organoid systems to find mechanisms and potential therapies for COVID19. I commend the authors for their thorough efforts in this manuscript, especially for being extremely diligent while preparing figures. This would definitely benefit the readers within and outside the field. I highly recommend this to be published in MDPI-Cells given the importance of the subject in the current pandemic context. The only suggestion from me would be to elaborate a bit more in the therapeutic implications (with a few sentences in the conclusion section) to highlight the possibility of isolating tissues samples and set up organoid systems from deceased COVID19 patients themselves to dig deeper into the mechanisms. This, I can understand, is extremely challenging given the contagious nature of the disease but may be an interesting aspect to speculate on the topic. Since long term lung damage is a potential effect of sever COVID19, identifying virus-host tissue interactions within a tissue sample that had been already infected with the virus in vivo may shed light on mechanisms of new treatments for the disease.

Author Response

Response to Reviewer 2 Comments (Please see the attachment)

To reviewer #2: We appreciate your careful reminding. Your comments are very valuable and helpful for revising our paper. All changes had been made with red color in the revised manuscript.

Point: I highly recommend this to be published in MDPI-Cells given the importance of the subject in the current pandemic context. The only suggestion from me would be to elaborate a bit more in the therapeutic implications (with a few sentences in the conclusion section) to highlight the possibility of isolating tissues samples and set up organoid systems from deceased COVID19 patients themselves to dig deeper into the mechanisms. This, I can understand, is extremely challenging given the contagious nature of the disease but may be an interesting aspect to speculate on the topic. Since long term lung damage is a potential effect of sever COVID19, identifying virus-host tissue interactions within a tissue sample that had been already infected with the virus in vivo may shed light on mechanisms of new treatments for the disease.

Response: Thank you for your advice. Your suggestion is valuable and We have added a few sentences in the conclusion section (as you suggested). The following parts marked in red are the added sentences in the revised manuscript.

“While isolating tissue samples and establishing lung organoids from deceased COVID-19 patients is challenging given the infectivity of SARS-CoV-2, this provides insight into virus-host interactions and may shed light on new therapeutic mechanisms for COVID-19.”

Reviewer 3 Report

The review authored by Peng et al. has done an excellent job of summarizing the use and applications of lung organoids to study SARS-CoV-2 infection. The background information on SARS-CoV-2 and lung organoids was appropriate. The authors have used the tables and figures to summarize the different lung organoid models and data on SARS-CoV-2 infection. The "studies on SARS-CoV-2 with lung organoids" section was well-written and covered most of the tested drugs in the lung organoid model. The conclusion was good and it is appreciable that the authors have also highlighted the pitfalls in the lung organoid models and have explained them clearly.

Overall, the manuscript is well drafted and appropriate comments were made from the literature. A comprehensive summary of the use of lung organoids in studying SARS-CoV-2 infection was made. Good job!

A few minor typo errors were found.

Kindly correct the following:

Title: Rewrite as "Lung Organoids as a model to study SARS-CoV-2 infection"

Line 124: Correct the typo "SAS-CoV-2" to "SARS-CoV-2"

Figure 1: Correct the typo "MeRS-CoV" to "MERS-CoV"

Author Response

Response to Reviewer 3 Comments (Please see the attachment)

To reviewer #3: We appreciate your careful reminding. Your comments are very valuable and helpful for revising our paper. All changes had been made with red color in the revised manuscript.

Points: A few minor typo errors were found, such as:

Title: Rewrite as "Lung Organoids as a model to study SARS-CoV-2 infection"

Line 124: Correct the typo "SAS-CoV-2" to "SARS-CoV-2"

Figure 1: Correct the typo "MeRS-CoV" to "MERS-CoV"

Responses: Thank you for your advice. We’ve checked the written errors throughout the manuscript and corrected the mistakes found in the revised manuscript.

Reviewer 4 Report

The authors present a comprehensive, instructive, and well-written review on the use of lung organoids for SARS-COV-2 studies. The review is ready for publication. The only item that should be addressed is that, in section 4.3, it is not clear how the lung organoids can be used for precision medicine. Do the authors refer to the possibility to establish patient-specific organoids from COVID-19 patients?

Author Response

Response to Reviewer 4 Comments (Please see the attachment)

To reviewer #4: We appreciate your careful reminding. Your comments are very valuable and helpful for revising our paper.

Point: In section 4.3, it is not clear how the lung organoids can be used for precision medicine. Do the authors refer to the possibility to establish patient-specific organoids from COVID-19 patients?

Response: Thank you for your advice. In this manuscript, we mentioned the possibility to establish patient-specific organoids from COVID-19 patients. In Section 4.3, we mainly described the application of lung organoids in precision medicine from the following two aspects: the first one is lung organoid culture combined with gene editing technology can achieve precision treatment (from line 258 to line 271); the second one is due to individual differences in the severity of COVID-19 disease, the establishment of patient-specific organoids from different patients (including different genders, ages, races, health conditions, etc.) can help to achieve personalized treatment (from line 272 to line 284).

All in all, thank you very much for your time and consideration. This revised manuscript has significantly been improved with your help. We hope this revision meets with your approval.

Round 2

Reviewer 1 Report

The  revised the manuscript is very much improved.  The authors replied to all of my comments.  I only have one comment. Since the authors replied to me that the lungs are one of the infected organs by the SARS-COV-2, and seriously damaged, I think the authors should write, "The lung is the major target organ infected and (seriously) damaged by SARS-CoV-2", instead of "The lung is the major target organ infected by SARS-CoV-2".

Author Response

Response to Reviewer 1 Comments

To reviewer #1: We appreciate your careful reminding. Your comments are very valuable and helpful for revising our paper. All changes had been made with red colored text in the current manuscript and our itemized responses to your comments (repeated below for your convenience) are as follows:

Point 1: The revised the manuscript is very much improved. The authors replied to all of my comments. I only have one comment. Since the authors replied to me that the lungs are one of the infected organs by the SARS-COV-2, and seriously damaged, I think the authors should write, "The lung is the major target organ infected and (seriously) damaged by SARS-CoV-2", instead of "The lung is the major target organ infected by SARS-CoV-2".

Response 1: Thank you for your comments. Your suggestion is very valuable and helpful. We have written “The lung is the major target organ infected and (seriously) damaged by SARS-CoV-2" instead of "The lung is the major target organ infected by SARS-CoV-2" in the abstract and also in the text (Line 63-).

All in all, thank you very much for your time and consideration. This revised manuscript has significantly been improved with your help. We hope this revision meets with your approval.
